# Probiotic Bacteria from Human Milk Can Alleviate Oral Bovine Casein Sensitization in Juvenile Wistar Rats

**DOI:** 10.3390/microorganisms11041030

**Published:** 2023-04-14

**Authors:** Kawtar Keddar, Hasnia Ziar, Noussaiba Belmadani, Magali Monnoye, Philippe Gérard, Ali Riazi

**Affiliations:** 1Laboratoire des Micro-Organismes Bénéfiques, des Aliments Fonctionnels et de la Santé (LMBAFS), Abdelhamid Ibn Badis University, Hocine Hamadou Street, Mostaganem 27000, Algeria; 2Laboratoire de Bio-Economie, Sécurité Alimentaire et Santé, Abdelhamid Ibn Badis University, Hocine Hamadou Street, Mostaganem 27000, Algeria; 3Micalis Institute, INRAE, AgroParisTech, Paris-Saclay University, 78350 Jouy-en-Josas, France

**Keywords:** probiotic, human milk, casein-induced allergy, Wistar rat, immunoglobulin E, inflammation

## Abstract

This study aims to see if probiotic bacteria from human milk could ameliorate oral cow’s milk sensitization. The probiotic potential of the SL42 strain isolated from the milk of a healthy young mother was first determined. Rats were then randomly gavaged with cow’s milk casein without an adjuvant or assigned to the control group. Each group was further subdivided into three groups, with each receiving only *Limosilactobacillus reuteri* DSM 17938, SL42, or a phosphate-buffered saline solution. Body weight, temperature, eosinophils, serum milk casein-specific IgE (CAS-IgE), histamine, and serum S100A8/A9 and inflammatory cytokine concentrations were measured. The animals were sacrificed after 59 days; histological sections were prepared, and the spleen or thymus weights, as well as the diversity of the gut microbiota, were measured. On days 1 and 59, SL42 abridged systemic allergic responses to casein by dropping histamine levels (25.7%), CAS-specific IgE levels (53.6%), eosinophil numbers (17%), S100A8/9 (18.7%), and cytokine concentrations (25.4–48.5%). Analyses of histological sections of the jejunum confirmed the protective effect of probiotic bacteria in the CAS-challenged groups. Lactic acid bacteria and Clostridia species were also increased in all probiotic-treated groups. These findings suggest that probiotics derived from human milk could be used to alleviate cow’s milk casein allergy.

## 1. Introduction

Food allergies are becoming more common around the world, particularly in developed countries, and are no longer a rare occurrence in Africa, where food allergies account for 5 to nearly 50% of allergic reactions [1]. In Algeria, food allergies affect 8.5% of schoolchildren, according to Yakhlef and Souiki [2]. Cow’s milk protein allergy (CMPA) is a type of food allergy that is most common in infants and children under the age of three. The most common symptoms are dermatitis, urticaria or oral allergy syndrome, and gastrointestinal (GI) disorders such as changes in stool frequency and consistency, mucous or blood spots in stools, infantile colic, nausea, vomiting, and gastroesophageal reflux [3]. CMPA affects approximately 8%of infants and young children. After egg, peanut, and fish allergies, it is the fourth-most common food allergy in babies [4].

A variety of factors may play a role in the complex development of food allergies. A disruption in the development of oral tolerance has been observed in infants with food allergies, characterized by defects in the induction of regulatory T cells, and the production of allergen-specific neutralizing IgA antibodies [4]. Furthermore, even though the specific properties of the allergens themselves contribute to the degree of allergenicity, defects in the epithelial barrier, both in the skin and in the intestine, as well as changes in the pH of the stomach are thought to promote allergy. Furthermore, we are only now beginning to understand how the microbiome can help with allergy problems [5,6].

Breastfeeding is undeniably beneficial, according to scientific evidence. Breast milk is the only milk that can be permanently adapted to the needs of a growing infant. One of the advantages of breastfeeding is that it helps to prevent allergies [7]. However, while a large number of studies support breastfeeding’s role in lowering the risk of allergy, other studies examining the effect of prolonged breastfeeding do not. Animal models and in vitro evidence indicate that the gut microbiome may protect against food allergy, and that probiotics may be a useful tool [8]. However, there is no consistent evidence for identifying the specific bacterial species, dosage, and optimal duration for achieving the desired immunomodulation [9]. Early-life probiotic supplementation via breast milk may be a useful approach to the primary prevention of a variety of human allergic diseases.

Breast milk from healthy women contains approximately 10^3^ to 10^4^ CFU/mL commensal microbes and serves as a source of probiotics for infants [10]. Human milk microbiota diversity is influenced by maternal and environmental factors such as breastfeeding practice, behavior, other milk components, genetics, geographical region, race, and population [11]. The evidence as to whether probiotics can induce tolerance in food allergy is currently not clear. To the best of our knowledge, published studies conducted on the anti-food allergy potential of probiotic bacteria gave variable results [12,13,14], and no one used probiotic strains from human milk. *Lactobacillus* strains, including the *L. casei* strain Shirota, the *L. plantarum* strain L-137, and the *L. acidophilus* strain L-92, have been reported as probiotics that modify antigen-specific serum immunoglobulin (IgE) levels in animal models [12,15,16]. We hypothesized that the prevention of allergy using probiotics may be more effective at juvenile age and that bacterial strains from human milk would be the most appropriate probiotics for that purpose. Therefore, in the present study, we assess the preventive effects of dietary intervention with probiotics from human milk to clarify their tolerogenic effect in managing food allergy symptoms at juvenile age. *Limosilactobacillus reuteri* Protectis (DSM 17938) or isolated SL42, whose genetic and probiotic properties were characterized, were given to juvenile female Wistar rats, sensitized intragastrically (i.g.) with casein from cow’s milk, as an animal model.

## 2. Materials and Methods

### 2.1. Bacteria Used in This Study

*Limosilactobacillus reuteri* Protectis DSM 17938 (formally known as *Lactobacillus reuteri*) was the reference strain from breast milk supplied by the PEDIACT laboratory BioGaia (Asnières-sur-Seine, France). SL42 is an isolated strain from the breast milk of a healthy young mother (Algeria). Each strain was grown in de Man, Rogosa, and Sharpe broth (Biomérieux, Craponne, France) supplemented with cysteine-HCl (MRS-cys) under anaerobic conditions at 37 °C for 24 h.

### 2.2. Using16S rRNA Gene Sequencing to Identify SL42 Isolate

Identification of SL42 was made primarily by partial sequencing of 16S rRNA genes. The extraction of bacterial genomic DNA was performed using the GF-1 Nucleic Acid Extraction Kit (Vivantis Technologies Sdn Bhd, Selangor DE, Malaysia) according to the manufacturer’s instructions. The complete 16S rRNA gene region was amplified via primers, 1492R (5′-GGTTACCTTGTTACGACTT-3′) and 27F (5′-AGAGTTTGATCCTGGCTCAG-3) (Vivantis Technologies Sdn Bhd, Selangor DE, Malaysia). The PCR products were further verified using 1% agarose gel electrophoresis and subjected to sequencing (Seri Kembangan, Selangor, Malaysia; https://apicalscientific.com/, accessed on 22 December 2022). The identification of the isolate was carried out by comparison with reference sequences using the NCBI BLAST algorithm (http://www.ncbi.nlm.nih.gov/blast, accessed on 22 January 2023). The neighbor-joining method was used to construct a phylogenetic tree (MEGA 6.0 program).

### 2.3. Characterization of the Probiotic Potential

To attribute the qualification of “probiotic” to the SL42 strain, the following tests were performed for both bacteria strains used in the in vivo part of this study, where the SL42 isolate was compared with the probiotic strain of DSM 17938 taken as reference. Bacterial loads were adjusted at 1 to 5 × 10^7^ CFU/mL in all experiments. All microbiological components were purchased from Merck and chemicals from Sigma (France), unless otherwise specified.

#### 2.3.1. pH and Bile Tolerance Assays

The method previously described by Ziar and Riazi [17] was slightly modified. Bacteria were cultivated (individually) into pH 2 MRS broths and incubated at 37 °C for 2 h. MRS-cys agar plates were used to determine viable counts every 30 min exposure. Bacteria bile salt tolerance was determined via the viable count method as previously described [18]. Following incubation for 24 h at 37 °C, the culture was centrifuged 5000× *g* at 40 °C for 10 min. Eventually, 0.02 mL of bacterial suspension was inoculated in freshly sterile phosphate buffer of pH 7.5 containing bile (0.3% *v*/*v*; Sigma-Aldrich, France). Following incubation at 37 °C for 24 h, viable counts were observed on MRS-cys agar plates.

#### 2.3.2. Detection of Antimicrobial Activity

Both probiotic strains were tested for antimicrobial activity. Seven pathogenic indicator bacteria and one fungus, *Candida albicans*, were used. Probiotic bacteria were cultured in an MRS-cys medium for 24 h at 37 °C, then centrifuged at 8000× *g* for 15 min at 4 °C (Thermo Scientific, Waltham, MA, USA), and cell-free supernatant yielded before the assay. The pH was adjusted to 6.5 with 6 mol/L NaOH, and then filtered via a membrane with a pore size of 0.22 μm. The modified Oxford cup double plate method was used to determine the antimicrobial activity [19]. Oxford cup (5 mm) was placed on an agar surface and the pathogen indicator exponential phase (100 μL, 1 × 10^7^ CFU/mL) was spread on the nutrient agar surface; then, 200 μL supernatant was added in wells. Following incubation at 30 °C for 24 h, the diameters of the clear inhibitory zone were measured.

#### 2.3.3. Hydrophobicity

Hydrophobicity of bacteria was determined using xylene extraction according to the method of Perez et al. [20], and hydrophobicity percentage (H %) was calculated using Equation (1):(1)H%=[(A0−A)/A0] 100
where *A*0 and *A* are absorbance values measured before and after xylene extraction.

#### 2.3.4. Hemolytic Activity

Bacterial cultures were observed on defibrinated sheep blood to a concentration of 5% (*w*/*v*) on blood agar plates [21], incubated for 24 h at 48 °C. Hemolytic activity was verified by β-hemolysis (bright zones around colonies), α-hemolysis (green zones around colonies), and γ-hemolysis (no zone around colonies) reactions.

#### 2.3.5. Cholesterol Uptake

The method of Ziar et al. [22] was followed. In brief, MRS-THIO medium containing 2% (*w*/*v*) sodium thioglycolate was supplemented with 85 μg/mL of water-soluble cholesterol, previously sterilized by filtration on a membrane Millipore (C1145, cholesterol–PEG 600; Sigma). Bile at a final concentration of 0.3% and the bacterial inoculum (1%, *w/v*) were then added. The milieu was incubated at 37 °C/24 h in anaerobic conditions (anaerobic jar with CO_2_ generating system, Anaerocult, Merckmillipore, Fontenay-sous-Bois, France). To estimate the amount of assimilated cholesterol, the cultures were centrifuged (2000× *g*/10 min at 4 °C) after 24 h of incubation, and the pellets were washed with sterile distilled water, and dried at 80 °C until the weight remained stable. Bacteria were tested for their cholesterol uptake capacities, expressed by the specific ability to reduce the available cholesterol from the culture medium after 24 h incubation, which was calculated according to the formula proposed by Pereira and Gibson [23].

#### 2.3.6. Antibiotic Susceptibility

An antibiotic susceptibility test was performed with 11 different antibiotics, including those used as cell wall or protein synthesis inhibitors, and the broad-spectrum antibiotics known to be effective against Gram-positive and Gram-negative bacteria [24]: amoxicillin, penicillin, gentamicin, streptomycin, chloramphenicol, norfloxacin, ciprofloxacin, sulfonamide, clindamycin, novobiocin, and vancomycin. Fresh overnight cultures of probiotic bacteria were inoculated separately at 0.5 McFarland in Mueller–Hinton agar. Subsequently, antibiotic discs were added onto inoculated Mueller–Hinton agar plates. The diameters of the inhibition zones around the antibiotic discs were observed following a 24–48 h incubation period at 37 °C. The sensitivity conditions were determined according to NCLS standards [25].

### 2.4. In Vivo Study

#### 2.4.1. Animal Housing

Female Wistar rats were shipped from Pasteur Institute (Algiers, Algeria) at 6 weeks of age (80–100 g). Housing rooms were kept at constant temperature (24 ± 2 °C) with an adequate light: dark cycle (12:12). All procedures were approved by the Animal Ethics Committee of the University of Mostaganem. Parental rats were housed in pairs in polycarbonate cages with *ad libitum* access to distilled and sterilized water and were fed a diet without allergens (SARL Aliment souris et rat: La Ration, Bouzaréah, Algiers, Algeria) for three successive generations. The third filial generation was used in this experiment at juvenile age and rats were acclimatized under the same conditions (2 rats/cage) cited above for 15 days before the beginning of the casein challenge.

#### 2.4.2. Experimental Design

The following in vivo experiment is illustrated in Figure 1. The female rats of 3 weeks old were randomly divided into 6 groups (n = 8 rats per group): control group receiving only phosphate-buffered saline solution (PBS) (C), nonsensitized group treated with SL42strain, nonsensitized group treated with DSM 17938 strain, casein-sensitized (Protifar©, 90% cow’s milk casein, EAN 8712400748124, NUTRICIA Nutrition Clinique, Rueil-Malmaison Cedex, France) group (CAS), casein-sensitized group treated with SL42 (CAS + SL42), and casein-sensitized group treated with DSM 19738 strain (CAS + DSM 19738).

Before (−3 day) and during casein challenge, each rat from the probiotic-treated groups received 10^8^CFU of DSM 19738 or SL42 in 1 mL physiological water by gavage every other day. Casein challenge was started by giving i.g. 60 mg casein without adjuvant during the first 42 days, then was associated to 20 mg gluten during the rest of the period of challenge (days 43–57). After that, sensitization using cow’s milk casein was interrupted for one day prior to sacrifice.

#### 2.4.3. Assessment of Macroscopic Casein Allergy Symptoms

Macroscopic casein allergy symptoms were assessed by monitoring rats every 30 min, and during the 3 h following cow’s milk casein sensitization. Clinical signs of anaphylaxis were scored depending on the gravity of the developed symptom: 0 for no signs; 1 for scratching and rubbing nose and head; 2 for bags around eyes and mouth; 3 for diarrhea; 4 for reduced activity with a satisfied respiratory rate, cyanosis around mouth and tail, and no activity; and 5 for death.

Diarrhea score was classified using Bristol scale: 1 for separate hard lump, 2 for lumpy feces, 3 for sausage-shaped feces with cracks on the surface, 4 for smooth and soft form, 5 for soft blobs with clear-cut edges, 6 for mushy blobs with ragged edges, and 7 for entirely liquid feces.

The weight of the rats was recorded during the experiment to assess the effect of CAS-induced sensitization on body weight. Rectal temperature was measured after cow’s milk casein sensitization and every 30 min (RET-2probe, Kent Scientific, Torrington, CT, USA). Stress status of rats was estimated by measuring serum uric acid concentration determined by fluorometry (MAK077-1KT, Sigma-Aldrich, Saint-Quentin-Fallavier, France).

#### 2.4.4. Determination of Specific Casein IgE, Histamine, S100A8/A9, Inflammation-Associated Cytokines, and Eosinophil Number

The levels of S100A8/A9, TLR4, NF-κB, TNF-α, IL-6, and IL-1β in the blood were determined using ELISA Assay Kit according to the manufacturer’s instructions (R & D Systems, Minneapolis, MN, USA). Serum was collected 30 min after casein administration and tested for histamine using an ELISA kit according to the manufacturer’s instructions (Beckman Coulter, Brea, CA, USA).

Serum levels of casein-specific IgE (CAS-IgE) were also assessed using ELISA kit (BD Biosciences, San Jose, CA, USA). The number of eosinophils in the blood was determined using Hemogram technique (Mindray 2800 BC brand automatic blood count device, Bath, UK).

#### 2.4.5. Cultivation of Bacteria from Feces

Fecal samples were collected before and after the challenge and plated on Hektoen/TSA for nonspecific bacteria, and MRS/MRS-cys media for lactic acid bacteria enumeration [26]. After 48–72 h incubation, the bacterial loads were expressed as CFU/g of fecal material. Clostridia species were counted after initial enrichment followed by plating on TSA II (4 days) with 5% sheep blood (Fisherscientific, CA) [27].

#### 2.4.6. Determination of Spleen/Body Weight Index and Thymus/Body Weight Index

The rats were weighed and sacrificed. The rats’ spleens and thymuses were removed and weighed immediately. The spleen index (SI) and thymus index (TI) were calculated using Equation (2) [28]:(2)SI or TI (mg/g)=(weight of spleen or thymus)/body weight

#### 2.4.7. Histological Analysis

Intestinal tissues (jejunum) were removed on the day of dissection, fixed in 10% phosphate-buffered formalin, and embedded in paraffin. The jejunum sections were stained with hematoxylin and eosin (HE) for the observation of inflammatory infiltrates and eosinophils and identification of goblet cells.

#### 2.4.8. Bacterial Translocation Test

The spleen and the mesenteric lymph nodes (MLN) of rats were macerated and suspended in sterile physiological saline. Serial dilutions were plated and incubated overnight at 37 °C on MacConkey agar (24 h), blood agar, and MRS-cys agar (48 h) (Merck, France).

### 2.5. Statistical Analysis

The mean ± standard error of the mean or standard deviation was used to present experiment’s data for statistical analysis implementation (SPSS statistics 26, Chicago, IL, USA). Statistical significance was determined using Student’s *t*-test, and one-way ANOVA was used for parametric tests. Differences at *p* < 0.05 were considered statistically significant.

### 2.6. Ethics Approval

The animal research presented in this manuscript was carried out ethically in accordance with the Helsinki Declaration and the ARRIVE guidelines for in vivo experiments. Furthermore, the Ethics Committee of the Faculty of Life Science and Nature affiliated with Abdelhamid Ibn Badis University of Mostaganem approved this research (Approval No. 2019-013).

## 3. Results

### 3.1. SL42 Is a Lacticaseibacillus rhamnosus as Confirmed by 16S rRNA Analysis

After catalase (negative), Gram tests (positive), and morphological (white and smooth colonies with approximately 2 mm diameter were picked from MRS-cys agar) and biochemical analyses, SL42 was subcultured at 37 °C to obtain pure cultures for molecular identification. The 16S rRNA gene sequence of strain SL42 was sequenced and compared against known strains based on BLAST searches. SL42 was subsequently identified as *Lacticaseibacillus rhamnosus* as confirmed by the results of a phylogenetic analysis (Figure 2). *L. rhamnosus* SL42 was deposited in NCBI GenBank under the accession number OQ300076, showing a sequence similarity of 98–99% when compared with the known *Lacticaseibacillus rhamnosus* species (Appendix A).

### 3.2. L. rhamnosus SL42 Expresses a Satisfying Probiotic Potential

The *L. rhamnosus* SL42 showed high acid tolerance and survivability at pH 2 (93%) after 2 h. The *L. rhamnosus* SL42 was also tolerant to pancreatic and pepsin enzymes under simulated digestive conditions (data not shown). Moreover, approximately 90.5% of *L. rhamnosus* SL42 cells survived with 0.3% bile and assimilated 6.01 mg/g cholesterol. The isolated *L. rhamnosus* SL42 strain also showed a high hydrophobicity of 51% (Table 1).

*L. rhamnosus* SL42 cells strongly inhibited *E. coli* and *Pseudomonas aeruginosa* with the highest inhibitory zones being 18 and 17 mm, respectively. The inhibitions of *Candida albicans, Staphylococcus aureus,* and *Klebsiella pneumoniae* were weaker, with inhibition zones between 11 and 15 mm (Table 2).

*L. rhamnosus* SL42 exhibited variable antibiotic susceptibility (5/11) to penicillin, amoxicillin, chloramphenicol, novobiocin, and norfloxacin (Table 3). However, the strain was found to be gentamicin-, sulfonamide-, and vancomycin-resistant (as was *L. reuteri* DSM 17938).

### 3.3. Macroscopic Symptoms Disappear after One-Week Casein Gavage

Diarrhea was only observed during the first week of CAS gavage. The diarrheic score was 7 and 4 on the Bristol scale for rats receiving exclusively CAS (50% of rats) or CAS-probiotic bacteria (33% of rats), respectively.

There were no differences in body weight and temperature between the experimental and the control groups (all *p* > 0.05) during the entire study (Appendix A). Uric acid levels in rat sera were significantly (all *p* < 0.05) increased from the 1st day to the 58th day in all CAS-sensitized rats (49.8%: CAS + SL42; 51.9%: CAS + DSM 17938; 74.7%: CAS) and remained unchanged in the control group and those receiving only individual probiotic bacteria (Appendix A).

### 3.4. Calprotectin, Eosinophils, and Cytokines Associated with CAS- Induced Allergy Were Successfully Decreased in Plasma of Rats Gavaged with the SL42 Strain

Significantly (*p* < 0.05) higher levels of CAS-specific IgE and histamine were detected in the sera of all rats treated with CAS. Sensitization with casein triggered the production of specific IgE with an average of 34.25 ± 1.25 (IU/L) in CAS-treated rats (Figure 3a). CAS-probiotic-treated rats exhibited an almost 50% reduction (*p* < 0.05) with registered values of 15.89 ± 0.89 IU/L on average in the SL42-treated group, and 17.98 ± 0.53 IU/L on average in the DSM 17938-treated group. Similar trends were obtained for histamine levels with25.6 ± 1.6 nmol/L (*p* < 0.05) in the CAS-treated group compared with 19 ± 1 and 20.5 ± 0.8 nmol/L for the SL4- and DSM 17938-treated groups, respectively (Figure 3b). The control groups and those receiving only individual probiotic bacteria produced neither CAS-specific IgE nor histamine.

The plasma levels of S100A8/A9 were increased in the CAS group compared with the control group and the probiotic-treated groups. The level of S100A8/A9 was statistically different on days 1 and 59 (all *p* < 0.05), as seen in Figure 4a. SL42 decreased calprotectin (S100A8/9) by 18.7% in CAS + SL42-treated rats.

The levels of TLR4 were higher in the CAS group than in the control group or the probiotic-treated groups, and significant differences were found on days 1 and 59 (all *p* < 0.05), as seen in Figure 4b. SL42 decreased TLR4 by 25.45% in CAS + SL42-treated rats.

The plasma levels of TNF-α, NF-κB, IL-1β, and IL-6 were higher in the CAS group than in the control group, and significant differences were found on day 1 and day 59 (all *p* < 0.05), as seen in Figure 4c–f, respectively. SL42 decreased TNF-α, NF-κB, IL-1β, and IL-6 by −14.31% to −48.58% in CAS + SL42-treated rats.

Moreover, the number of eosinophils was also increased, with an average of 108.5 ± 10.66/mm^3^ in CAS-sensitized rats without probiotic treatment (Figure 5). Administration of the SL42 strain decreased the number to 90 ± 1.41/mm^3^ (−16.67%), whereas treatment with the probiotic strain DSM17938decreased eosinophil numbers to 93 ± 7.03/mm^3^ (−13.88%). Groups receiving only the individual probiotic bacteria exhibited ~50% lower values, ranging from 51.25 to 52/mm^3^ (Figure 5).

### 3.5. Probiotic Administration Modifies LAB and Clostridia Populations in Rats

Figure 6 shows the population of lactic acid bacteria (LAB) and nonspecific bacteria obtained by plating techniques. Before the CAS challenge (data not shown), the bacterial profile was comparable (*p* > 0.05) between nonsensitized and CAS-sensitized rats. Administration of probiotic bacteria markedly increased the density of LAB (Figure 6a) and Clostridia species, but not of nonspecific bacteria (Figure 6b). After the 58th day of the CAS challenge, the rats exhibited a diminished density of fecal LAB and an increased density of nonspecific bacteria (Figure 6a,b). The alterations in the enteric bacteria observed in CAS-treated rats were restored by the administration of SL42, where the populations of LAB (Figure 6a) and nonspecific microbes were significantly (Figure 6b) (*p* < 0.05) increased and decreased, respectively.

### 3.6. Probiotic Administration Does Not Change Spleen and Thymus Weights

Figure 7 shows the effects of the two probiotic strains SL42 and DSM 17938 on the thymus and spleen indices of the rats. Compared with the control group, the thymus and spleen indices were not found to be significantly different in the SL42 and DSM 17938 probiotic-treated groups (all *p* > 0.05).

### 3.7. Inflammation of Jejunal Tissue and Eosinophil Infiltration were Significantly Reduced by Probiotic Treatment

Hematoxylin and eosin staining showed that the jejunal mucosa was inflamed in the CAS sensitization group (Figure 8). The histological inflammation score in that group was2 (mild to moderate), and eosinophil infiltration was also significantly increased (*p* < 0.05) compared with the control (Figure 8). In contrast, the intestinal inflammation scores and degrees of eosinophil infiltration were significantly reduced by probiotic treatment in comparison with the CAS sensitization group (casein) (Figure 8). Moreover, villus length in the casein group was significantly reduced (*p* < 0.05), although the probiotic treatment groups showed almost normal features similar to those in the controls (Figure 8).

### 3.8. Probiotic Bacteria SL42 and DSM17938 Prevent Bacterial Translocation to Mesenteric Lymph Nodes in Wistar Rats Sensitized with Casein

The mesenteric lymph nodes were sterile in the control and probiotic groups while casein sensitization caused bacterial translocation (*p* < 0.05). Probiotic gavage completely eliminated bacterial translocation to MLN in rat groups subjected to the casein challenge (Table 4).

## 4. Discussion

The most common food allergy in children is cow’s milk allergy. There is currently no effective treatment available to prevent or cure food allergies [4]. However, according to numerous studies, breastfeeding protects the infant from developing allergic diseases, helps young infants’ immune systems mature, and protects them from infections [4,29].

There are many immunological components in human milk, such as probiotic bacteria, nondigestible oligosaccharides, secretory IgA, mucins, cytokines, long-chain PUFA, and hormones. Probiotic bacteria, in particular, may support immunocompetence, which is required for adequate capacity to induce oral tolerance, either directly or indirectly through stimulation of beneficial intestinal microbiota [30,31]. The benefits of breastfeeding to newborn health are meaningful, and the microbiome in milk may play a crucial role. In this context, we aimed to investigate the beneficial effects of probiotic bacteria found in human milk that may be associated with improved infant health, and could be incorporated in human milk formula. The goal of this study was to compare the effects of probiotic strains from human milk supplementation on the outcome of the allergic response in rats during oral sensitization with bovine casein. First, the strain SL42 isolated from the breast milk of a young and healthy mother was assessed for probiotic aptitudes and compared with the probiotic strain of *Limosilactobacillus reuteri* DSM 17938. The 16S rRNA analysis showed that the isolate belongs to the *Lacticaseibacillus rhamnosus* species. Kang et al. [32] also obtained two *Lacticaseibacillus rhamnosus* strains from the breast milk of healthy Chinese women, and this bacterial species is known to be one of the most prevalent bacterial species in human milk.

*Streptococcaceae*, *Pseudomonadaceae*, *Staphylococcaceae*, *Lactobacillaceae*, and *Oxalobacteraceae* are the common bacterial families [33]. In our study, the isolated strain *Lacticaseibacillus rhamnosus* SL42 performed better than DSM 17938, including by having better tolerance to acidity and bile, and antimicrobial ability. For antibiotic susceptibility, both assayed strains showed similar trends, especially for vancomycin resistance, as was previously observed for *Lacticaseibacillus rhamnosus* and *Limosilactobacillus reuteri* species in several studies [34,35]. In general, our isolated SL42 strain passed all the tests to be considered as a safe, well-tolerated, and efficacious probiotic-like strain that is able to contribute to beneficial effects on gut health.

It is known that Lactobacilli are an important part of normal human microbial flora that commonly colonize the mouth, the gastrointestinal tract, and the female genitourinary tract [36]. The scientific community agrees on the importance of strain specificity in the action of probiotic microorganisms on the health of their hosts. According to Xavier-Santos et al. [37], in addition to daily doses, researchers must consider the multiple action mechanisms that are unique to each species/strain.

After being identified, both strains of SL42 and DSM 17938 were included individually in our in vivo casein-induced allergy study. *L. reuteri* DSM 17938 was chosen because it is already delivered as a drug to children for alleviating gastrointestinal symptoms. Numerous clinical studies have suggested that *L. reuteri* may be beneficial in modulating gut microbiota, thereby eliminating infections such as enteric colitis, antibiotic-associated diarrhea, *Helicobacter pylori* infection, irritable bowel syndrome, inflammatory bowel disease, and chronic constipation. *L. reuteri* reduces the duration of acute infectious diarrhea in both children and adults and relieves abdominal pain in patients with colitis or inflammatory bowel disease [10,36].

To define the proper food allergy model, we proposed an oral sensitization model without an adjuvant that mimicked what happens in humans by using female Wistar rats of juvenile age. We believe that our model is appropriate because the administration of an adjuvant may influence the IgE response or cause a false-positive IgE response with a non allergenic food [37]. *Lacticaseibacillus rhamnosus* SL42 or *L. reuteri* DSM 17938 were given to Wistar rats at 3 weeks of age and the rats were challenged orally with casein. In this second part, macroscopic symptoms after casein gavage, calprotectin, eosinophils, and cytokine-associated CAS-induced allergy, fecal bacteria enumeration, changes in spleen and thymus weights, jejunal tissue and eosinophil infiltration, and bacterial translocation to mesenteric lymph nodes were all examined. During the sensitization period, all rats appeared healthy with similar weights between all the groups. Furthermore, no severe symptoms, such as death, were observed. One rat in the group sensitized only with casein had a score of 1, six had a score of 2, and one had a score of 3. There were no abnormalities in either the control or probiotic-treated groups. This could indicate that the model proposed herein is mildly allergic, and that, despite causing mild intestinal inflammation, it may have a long-term impact on rat growth and development. In general, both the isolated SL42 strain and *L. reuteri* DSM 17938 acted similarly in vivo.

Stanojevic et al. [29] revealed that early postnatal treatment with *Lactobacillus rhamnosus* LB64 appears to be effective in attenuating TNBS autoimmune encephalomyelitis. Similarly, early colonization with *L. rhamnosus* GG increased the richness and diversity of the colonic microbiota and promoted epithelial cell proliferation, differentiation, and mucosal IgA production in adults [38]. Torii et al. [12] found that *L. acidophilus* L-92 administration inhibited total IgE and OVA-specific IgE production in both in vivo and in vitro studies. Based on their findings, the authors hypothesize that LAB suppresses IgE production via a mechanism other than a shift to Th1-dominant immunity.

In our study, oral administration of probiotic strains from human milk in CAS-sensitized rats could reduce symptom scores, CAS-specific IgE, calprotectin, allergen-specific cytokines, and histamine release levels. Although no specific mechanism could be determined based on these data, our principal aim was to assess the direct role of these strains, and our results seem to be in line with the literature confirming the possibility of alleviating allergy markers through probiotic administration. Neau et al. [39] described the protective effect of the *Lactobacillus salivarius* LA307 strain on sensitization, with a decrease in allergen-specific IgE and allergy. In addition to those findings, Esber et al. [40] demonstrated in mice that giving *Lactobacillus rhamnosus* LA305, *L. salivarius* LA307, or *Bifidobacterium longum* subsp. *infantis* LA308 for 3 weeks after sensitization and challenge altered the composition of the gut microbiota. Cytokine production was significantly reduced by all probiotic strains. According to the authors, the three probiotic strains tested alter immune responses by inducing tolerogenic allergies and anti-inflammatory responses.

S100A8/A9 (calprotectin) is claimed to be a sensitive biomarker for inflammatory diseases such as rheumatoid arthritis, psoriasis, and vasculitis [41]. According to Zhu et al. [42], calprotectin, along with other inflammatory factors, may promote the inflammation seen in mild food allergies. S100A8/A9 is involved in innate immune responses in Baker’s asthma pathogenesis and is regulated by TLR4 polymorphisms [43].

We also found that CAS sensitization alone changed the composition of gut microbiota in comparison with the controls, in terms of the relative abundance of LAB, nonspecific bacteria, and *Clostridia* species. Our probiotic bacteria intervention was able to restore beneficial microflora in all probiotic-treated rats. Similarly, Tulyeu et al. [31] recently reported that allergen immunization in a food allergy model induced profound changes in the composition of the gut microbiome. The impact on gut microbiota is a proof of concept in this study, even though rat microbiota cannot be compared to human gut microbiota. However, we believe that several changes in the microbiota caused by the SL42 strain may contribute to or enhance its protective effect. Many anti-inflammatory properties have been reported for *L. reuteri* DSM 17938 in the literature. It generates reuterin, a powerful antimicrobial compound capable of inhibiting the growth of Gram-positive and Gram-negative bacteria, fungi, and protozoa [36]. Furthermore, *L. reuteri* forms a probiotic-rich biofilm, inhibits the production of proinflammatory cytokines, and prevents intestinal overgrowth by other commensals, thereby maintaining a balanced gut environment [36].

The “leaky gut syndrome” and bacterial translocation are considered by some authors as triggering factors for the onset of the disease as they promote chronic systemic inflammation. The most reported health benefits were from oral probiotic administration and fecal microbial transplantation [44]. Therapies that focus on modulating the gut microbiota are a good option for pediatrics, especially because infants have developing microbial communities that are associated with immune system maturation [37]. Interestingly, the two tested probiotic strains were able to abolish bacterial translocation in our allergy model, suggesting that the beneficial effects may be due to gut barrier reinforcement.

## 5. Conclusions

Human milk is an excellent source of LAB strains, which are commonly used as probiotics. *Lacticaseibacillus rhamnosus* strain SL42 was isolated from the breast milk of healthy Algerian women. Its probiotic potential was assessed in vitro using *L. reuteri* Protectis DSM 17938 as the reference strain. In summary, our findings show that supplementing juvenile rats with *L. rhamnosus* SL42 induces tolerogenic responses and serves several purposes, from lowering the level of casein-associated allergy parameters to improving macroscopic symptoms and suppressing bacterial translocation to MLN. Its effects were similar to those expressed by the probiotic strain of *L. reuteri* DSM 17938. This research identified a potential probiotic candidate for use in the food and pharmaceutical industries. Clinical studies will be required to confirm these experimental findings.

## Figures and Tables

**Figure 1 microorganisms-11-01030-f001:**
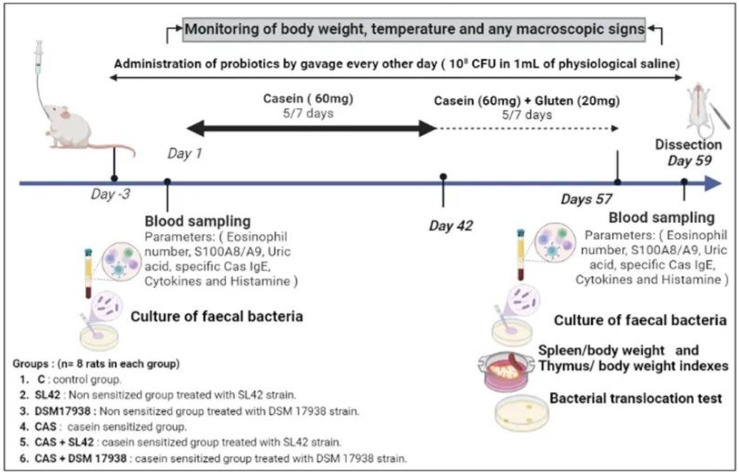
Experimental design of the present study. Bovine casein (Protifar©) was taken as allergen and was administered intragastrically to rats without the use of adjuvant. A total of 48 female rats (3 weeks old) were randomly divided into 6 groups (n = 8 rats per group): control group receiving only PBS (C group), nonsensitized group treated with SL42 strain (SL42 group), nonsensitized group treated with DSM 17938 strain (DSM 1938 group), casein-sensitized group (CAS group), casein-sensitized group treated with SL42 (CAS + SL42 group), and casein-sensitized group treated with DSM 19738 strain (CAS- + DSM 19738 group). In all probiotic-treated groups, SL42 or DSM 17938 were given every other day from day −3 to day 58.

**Figure 2 microorganisms-11-01030-f002:**
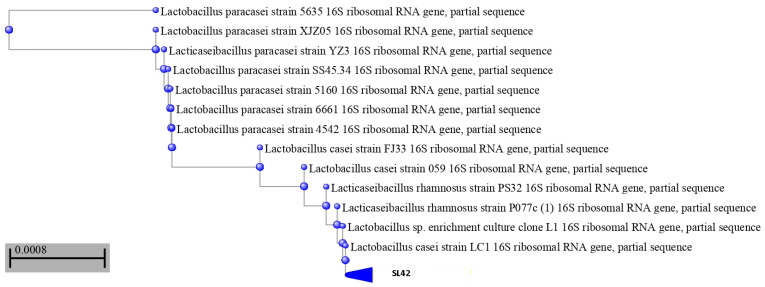
Constructed phylogenetic tree showing the position of the isolated SL42 with related *Lactobacillus* species.

**Figure 3 microorganisms-11-01030-f003:**
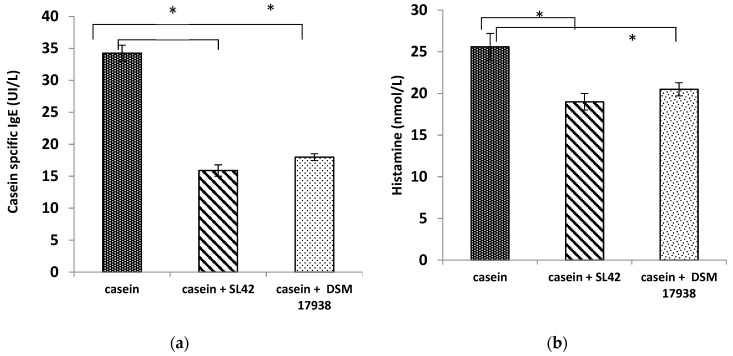
CAS-IgE (**a**) and histamine levels (**b**) in sera assessed using ELISA (n = 8 rats/ group). The data are presented as the mean ± SEM. Statistical analysis was conducted by using one-way ANOVA with Tukey’s multiple comparisons test. * *p* < 0.05. Casein-sensitized group receiving only casein (casein group), casein-sensitized group treated with SL42 (casein + SL42 group), and casein-sensitized group treated with DSM 17938 strain (casein- + DSM 17938 group).

**Figure 4 microorganisms-11-01030-f004:**
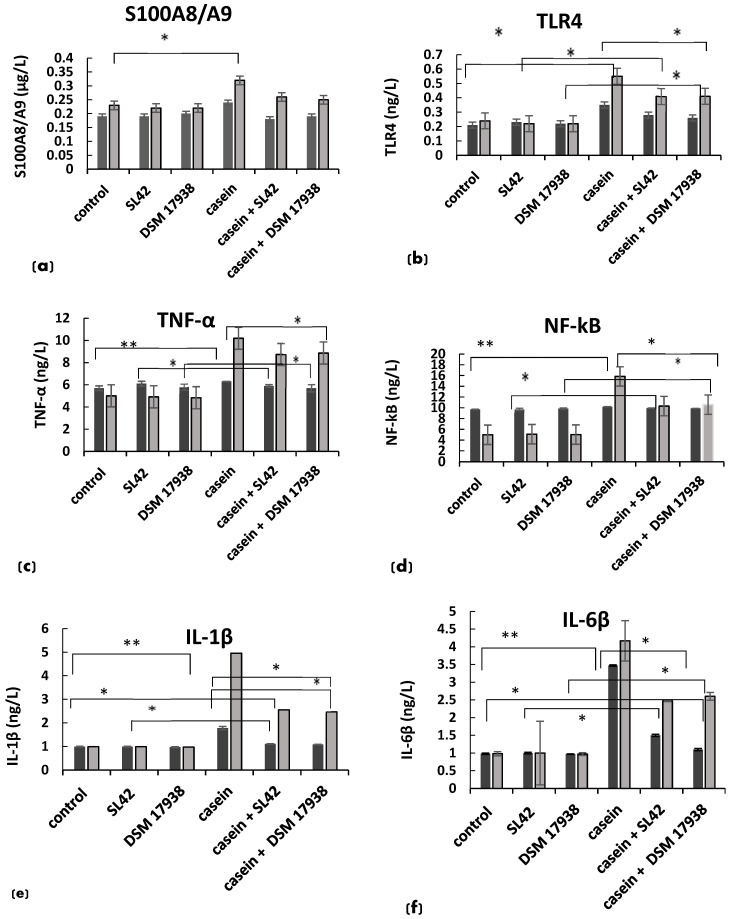
The expression of S100A8/A9 and associated cytokines in Wistar rats sensitized intragastrically by administration of casein without adjuvant (before 1st day, after 58th day). S100A8/A9 (**a**), TLR4 (**b**), TNF-α (**c**), NF-κB (**d**), IL-1β (**e**), and IL-6 (**f**) levels in sera of rats from different groups are shown. Values are means ± SEM (n = 8 rats/group). * *p* < 0.05. ** *p* < 0.01. Control group receiving only PBS (control group), nonsensitized group treated with SL42 strain (SL42 group), nonsensitized group treated with DSM 17938 strain (DSM 17938 group), casein-sensitized group (casein group), casein-sensitized group treated with SL42 (casein + SL42 group), and casein-sensitized group treated with DSM 17938 strain (casein- + DSM 17938 group).

**Figure 5 microorganisms-11-01030-f005:**
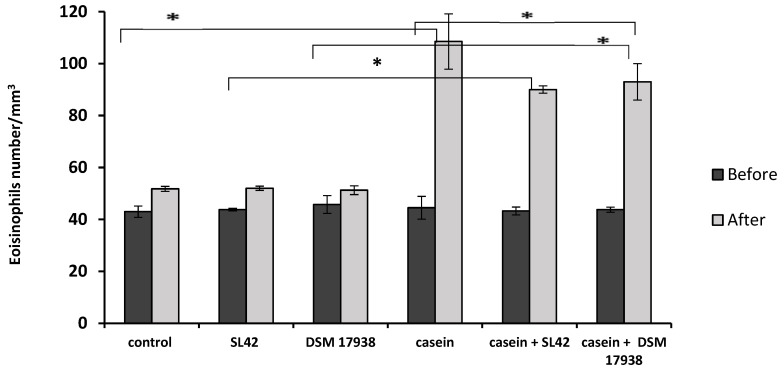
Eosinophil numbers in blood as determined using hemogram technique (n = 8 rats/ group). Wistar rats were sensitized intragastrically by administration of casein without adjuvant (before 1st day, after 58th day). Data are presented as the mean ± SD. Statistical analysis was conducted by using one-way ANOVA with Tukey’s multiple comparisons test. * *p* < 0.05. Control group receiving only PBS (control group); nonsensitized group treated with SL42 strain (SL42 group); nonsensitized group treated with DSM 17938 strain (DSM 17938 group); casein-sensitized group (casein group); casein-sensitized group treated with SL42 (casein + SL42 group); and casein-sensitized group treated with DSM 17938 strain (casein- + DSM 17938 group).

**Figure 6 microorganisms-11-01030-f006:**
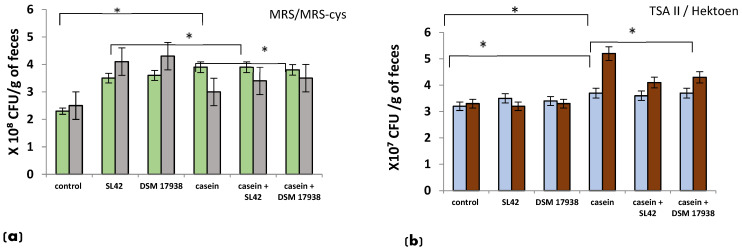
Bacteria numbers as determined by plating technique (n = 8 rats/ group). Lactic acid bacteria (**a**), Clostridia species (**b**). Wistar rats were sensitized intragastrically by administration of casein without adjuvant. The rats were fed every other day with *L. rhamnosus* SL42 or *L. reuteri* DSM 17938 at 1 × 10^8^ CFU/mL (feces were collected on the 59th day). The dose volume was 1 mL. The control group was fed with sterilized PBS solution. The data (MRS 
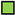
, MRS-cys 
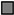
, TSA II 
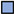
, Hektoen 
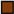
) are presented as the mean ± SD. Statistical analysis was conducted by using one-way ANOVA with Tukey’s multiple comparisons test. * *p* < 0.05. Control group receiving only PBS (control group); nonsensitized group treated with SL42 strain (SL42 group); nonsensitized group treated with DSM 17938 strain (DSM 17938 group); casein-sensitized group (casein group); casein-sensitized group treated with SL42 (casein + SL42 group); and casein-sensitized group treated with DSM 17938 strain (casein- + DSM 17938 group).

**Figure 7 microorganisms-11-01030-f007:**
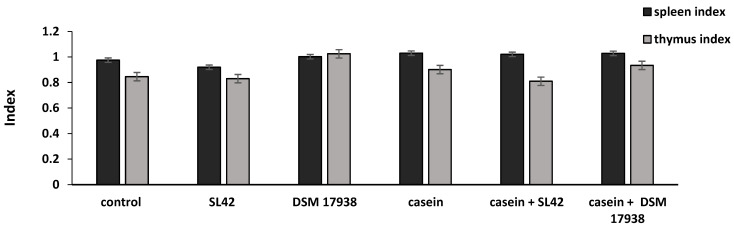
Effects of probiotic bacteria on the thymus and spleen indices of rats. Wistar rats were sensitized intragastrically by administration of casein without adjuvant. The rats were fed every other day with *L. rhamnosus* SL42 or *L. reuteri* DSM 17938 at 1 × 10^8^ CFU/mL. The dose volume was 1 mL. The control group was fed with sterilized PBS solution. Thymus and spleen samples from each group were collected on the 59th day (day of sacrifice). The thymus and spleen indices were measured as the ratio of the thymus or spleen weight to rat body weight. Values are means ± SD (n = 8 rats/group). No significant differences were observed at *p* > 0.05. Control group receiving only PBS (control); nonsensitized group treated with SL42 strain (SL42); nonsensitized group treated with DSM 17938 strain (DSM 17938); casein-sensitized group (casein); casein-sensitized group treated with SL42 (casein + SL42); and casein-sensitized group treated with DSM 17938 strain (casein- + DSM 17938).

**Figure 8 microorganisms-11-01030-f008:**
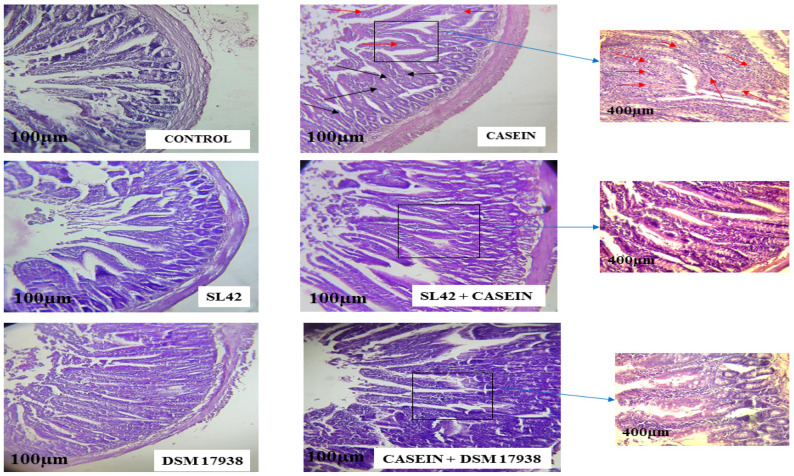
Representative HE-stained sections of jejunal mucosae are shown at original magnification, 100 µm. Wistar rats were sensitized intragastrically by administration of casein without adjuvant. The rats were fed every other day (−3rd to 58th day) with *L. rhamnosus* SL42 or *L. reuteri* DSM 17938 at 1 × 10^8^ CFU/mL. The dose volume was 1 mL. The black and red arrows indicate eosinophil infiltration and goblet cells, respectively. Control group receiving only PBS (control); casein-sensitized group (casein); nonsensitized group treated with SL42 strain (SL42); casein-sensitized group treated with SL42 (Casein + SL42); nonsensitized group treated with DSM 17938 strain (DSM 17938); and casein-sensitized group treated with DSM 17938 strain (Casein- + DSM 17938).

**Table 1 microorganisms-11-01030-t001:** Probiotic characteristics * of SL42 strain reported in this study.

*L. rhamnosus* SL42		*L. reuteri* DSM 17938
Parameters	Cell Viability/Activity	
Acid resistance (pH = 2)	2 × 10^7^ CFU/mL (0 min)	5 × 10^7^ CFU/mL (0 min)
	7.5 × 10^6^ CFU/mL (30 min)	2.3 × 10^7^ CFU/mL (30 min)
	6.9 × 10^6^ CFU/ mL (60 min)	1.9 × 10^6^ CFU/mL (60 min)
	6.6 × 10^6^ CFU/mL (120 min)	1.2 × 10^6^ CFU/mL (120 min)
Bile survival (0.3%)	5 × 10^7^ CFU/mL (0 min)	2.3 × 10^7^ CFU/mL (0 min)
	1.4 × 10^7^ CFU/mL (4 h)	2.1 × 10^6^ CFU/mL (4 h)
	9.3 × 10^6^ CFU/mL (24 h)	1.2 × 10^6^ CFU/mL (24 h)
Cholesterol uptake	6.01 mg/g	6.09 mg/g
Hydrophobicity (%)	51%	76%
Hemolytic activity	γ-hemolytic (no hemolysis)	γ-hemolytic (no hemolysis)

* The results are means of 3 independent replicates (n = 3).

**Table 2 microorganisms-11-01030-t002:** Antimicrobial activity * of probiotic bacteria.

Pathogens	Inhibition Zone (mm)	
	*L. rhamnosus* SL42	*L. reuteri* DSM 17938
*Candida albicans* ATCC 10231	14 ± 0.18	13 ± 0.36
*Escherichia coli* ATCC 25922	18 ± 0.02	14 ± 0.10
*Bacillus cereus* ATCC 10876	9 ± 0.11	6± 0.10
*Staphylococcus aureus* ATCC 33862	15 ± 0.30	8 ± 0.20
*Pseudomonas aeruginosa* ATCC 27853	17 ± 0.22	15 ± 0.20
*Salmonella enterica* subsp. *enterica* serotype *Enteritidis* ATCC 4931	9 ± 0.08	16 ± 0.04
*Klebsiella pneumoniae* ATCC 13883	11 ± 0.02	14 ± 0.20
*Shigella* sp. (isolate from our collection)	5 ± 0.02	6 ± 0.10

* The results are means of 3 independent replicates (n = 3) ±SD.

**Table 3 microorganisms-11-01030-t003:** Antibiotic susceptibility * of probiotic bacteria.

Probiotic Bacteria	Antibiotic Susceptibility (mm)										
	Amoxicillin (30 µg)	Ciprofloxacin (5 µg)	Gentamicin (10 µg)	Penicillin (10 µg)	Sulfonamide (25 µg)	Streptomycin (10 µg)	Clindamycin (2 µg)	Chloramphenicol (30 µg)	Vancomycin (30 µg)	Norfloxacin (10 µg)	Novobiocin (4 µg)
*L. rhamnosus* SL42	15 ± 0.1 I	0 ± 0.01 R	10 ± 0.02 R	20 ± 0.01 S	0 ± 0.1 R	0 ± 0.1 R	0 ± 0.0 R	25 ± 0.1 S	8 ± 0.03 R	16 ± 0.1 I	22 ± 0.02 S
*L. reuteri* DSM 17938	0 ± 0.01 R	16 ± 0.02 I	8 ± 0.01 R	15 ± 0.01 I	0 ± 0.1 R	16 ± 0.1 I	18 ± 0.0 I	16 ± 0.1 I	12 ± 0.02 R	16 ± 0.1 I	22 ± 0.0 S

* The results are means of 3 independent replicates (n = 3) ±SD. Abbreviations are S, sensitive; I, intermediate; R, resistant. The zone of inhibition (mm) for each antibiotic was measured and interpreted as follows: ≥19 mm = sensitive (S); 15–18 mm = intermediate (I); ≤14 mm = resistant (R).

**Table 4 microorganisms-11-01030-t004:** Bacterial translocation to mesenteric lymph nodes in rats subjected to casein challenge.

Group	Number of Affected Rats	CFU/ MLN of Rat
Control	0/8	0
SL42	0/8	0
DSM17938	0/8	0
Casein	6/8	288 *
Casein + SL42	0/8	0
Casein + DSM17938	0/8	0

Bacterial translocation data are represented as mean of the total CFU cultured from MLN of each rat (n = 8) after 48 h of incubation. * *p* < 0.05 compared with controls.

## Data Availability

Not applicable.

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
