# Peer review of "Probiotic Bacteria from Human Milk Can Alleviate Oral Bovine Casein Sensitization in Juvenile Wistar Rats"

_microorganisms, 2023, doi:10.3390/microorganisms11041030_

Round 1

Reviewer 1 Report

The paper entitled “Probiotic bacteria from human milk can alleviate oral’s bovine casein sensitization in juvenile Wistar rats.” Summarized the effect of probiotic bacteria isolated from human milk could ameliorate oral cow's milk sensitization. The authors obtain a probiotic bacterial strain identified as Lacticaseibacillus rhamnosus SL42 using 16s rRNA. The authors inoculated this probiotic bacterial strain into juvenile female Wistar rats and measured different parameters. The manuscript contains promising data but needs major revision before being considered for publication in microorganisms.   

1-    The abstract should be rephrased to contain the real data.

2-    Line 80, “16. S rRNA” should be “16S rRNA”

3-    Line 108 “Probiotic acteria” should be “Probiotic bacteria”

4-    The number of titles and subtitles should be checked throughout the manuscript.

5-    Lines 246 and 247, “Lacticaseibacillus rhamnosus” must be in italic. Please correct this throughout the manuscript.

6-    Figure S1 must be reorganized to be clear.

7-    I recommend transferring Table 1 to supplementary data and transferring Figure S1 (phylogenetic tree) to the main text.

8-    In table 3, some clear zone recorded 4, 6, and 9 mm, please specify the diameter of cup or well that is filled with cell-free filtrate.

9-    The footnote of the table “*The results are means of three independent experiments.” Should be replaced by “*The results are means of three independent replicates (n = 3) ±SD (or SE, please specify).

10- In Table 4, please refer to the meaning of S and R and their diameter of the clear zone in the table footnote.

11- In lines 426, 433, 437, 438, the scientific names must be in italic, please check and revised throughout the manuscript.

12-  The conclusion should be rephrased to summarize promising data and to contain the overall conclusion.

Author Response

First of all, we would like to thank you for the reviewer’s detailed comments and the constructive criticism that has helped to improve the manuscript. A marked copy of the manuscript has been uploaded for the reviewers and editors. We took into account all these comments for the revision of our manuscript and we believe that it significantly strengthened our manuscript. All authors have read and agreed to the revised version of the manuscript.

In the responses to reviewers file, we have included the original comments from the three referees and we developed a point-by-point response in red italics to each of the reviewer’s comment as expected by the journal Editorial committee. Therefore, we believe that we addressed the different issues which were raised by the reviewers.

In response to comments by Reviewer 1

Reviewer 1

Comments and Suggestions for Authors

The paper entitled “Probiotic bacteria from human milk can alleviate oral’s bovine casein sensitization in juvenile Wistar rats.” Summarized the effect of probiotic bacteria isolated from human milk could ameliorate oral cow's milk sensitization. The authors obtain a probiotic bacterial strain identified as Lacticaseibacillus rhamnosus SL42 using 16s rRNA. The authors inoculated this probiotic bacterial strain into juvenile female Wistar rats and measured different parameters. The manuscript contains promising data but needs major revision before being considered for publication in microorganisms.   

1-    The abstract should be rephrased to contain the real data.

This has been corrected in the revised version.

Rephrase:

The study aims to see if probiotic bacteria from human milk could ameliorate oral cow's milk sensitization. The probiotic potential of the SL42 isolate from the milk of a healthy young mother was first determined. The rats were then randomly gavaged with cow's milk casein without adjuvant or assigned to the control group. Each group was further divided into three, with each receiving only Limosilactobacillus reuteri DSM 17938, SL42, or a phosphate buffered-saline solution. Body weight, temperature, eosinophils, serum milk casein-specific IgE (CAS-IgE), histamine, and serum S100A8/A9 and inflammatory cytokines concentrations were all measured. The animals were sacrificed after 59 days; histological sections were prepared, and the spleen or thymus weights, as well as the diversity of the gut microbiota, were measured. On days 1 and 59, SL42 abridged systemic allergic responses to casein by dropping histamine (25.7%), CAS-specific IgE levels (53.6%), eosinophil number (17%), S100A8/A9 (18.7%), and cytokine concentrations (25.4–48.5%). Analyses of histological sections of jejunum confirmed the protective effect of probiotic bacteria in the CAS-challenged groups. Lactic acid bacteria and Clostridia species were also increased in all probiotic-treated groups. These findings suggest that probiotics derived from human milk could be used to alleviate cow's casein milk allergy.

2-    Line 80, “16. S rRNA” should be “16S rRNA”

This has been corrected in the revised version.

3-    Line 108 “Probiotic acteria” should be “Probiotic bacteria”

This has been corrected in the revised version.

4-    The number of titles and subtitles should be checked throughout the manuscript.

This has been checked and corrected in the revised version.

5-    Lines 246 and 247, “Lacticaseibacillus rhamnosus” must be in italic. Please correct this throughout the manuscript.

It is in italic in the Word version. Pdf conversion seems to change the italic format.

6-    Figure S1 must be reorganized to be clear.

This has been corrected in the revised version.

7-    I recommend transferring Table 1 to supplementary data and transferring Figure S1 (phylogenetic tree) to the main text.

This has been corrected in the revised version.

8-    In table 3, some clear zone recorded 4, 6, and 9 mm, please specify the diameter of cup or well that is filled with cell-free filtrate.

Sterile Oxford cups (5 mm) were placed on the surface of plates. The inhibitory activity was evaluated by measuring the diameter of the transparent inhibition zone.

9-    The footnote of the table “*The results are means of three independent experiments.” Should be replaced by “*The results are means of three independent replicates (n = 3) ±SD

This has been corrected in the revised version.

(or SE, please specify).

SEM for all Elisa results, SD for the rest

10- In Table 4, please refer to the meaning of S and R and their diameter of the clear zone in the table footnote.

This has been corrected in the revised version

11- In lines 426, 433, 437, 438, the scientific names must be in italic, please check and revised throughout the manuscript.

This has been corrected in the revised version

12-  The conclusion should be rephrased to summarize promising data and to contain the overall conclusion.

This has been corrected in the revised version

Reviewer 2 Report

In manuscript "Probiotic bacteria from human milk can alleviate oral’s bovine  casein sensitization in juvenile Wistar rats" detal description and characterisation of potentially probiotic strain Limosilactobacillus reuteri as well as its effect on casein sensitization in juvenile rats are monitored.

The methods are clearly written and the results are properly presented and the authors draw realistic conclusions from them. In my opinion, an original and very interesting manuscript. Apart from minor mistakes throughout the text, I have no comments.

It should be noted throughout the text that Latin names are in italics.

Author Response

Comments and Suggestions for Authors

In manuscript "Probiotic bacteria from human milk can alleviate oral’s bovine  casein sensitization in juvenile Wistar rats" detal description and characterisation of potentially probiotic strain Limosilactobacillus reuteri as well as its effect on casein sensitization in juvenile rats are monitored.

The methods are clearly written and the results are properly presented and the authors draw realistic conclusions from them. In my opinion, an original and very interesting manuscript. Apart from minor mistakes throughout the text, I have no comments.

We thank Reviewer 2 for this very positive evaluation of our manuscript

It should be noted throughout the text that Latin names are in italics.

This has been corrected in the revised version

Reviewer 3 Report

The manuscript "Probiotic bacteria from human milk can alleviate oral’s bovine casein sensitization in juvenile Wistar rats" has an interesting result. However, for publication in Microorganisms, the manuscript needs to be improved.

The discussion needs to review and compare the data with the literature. Many grammatically problematic sentences were found throughout the manuscript, which must be checked and corrected precisely.

1.     L15: Use the full form of all abbreviated words for the first time throughout the manuscript

2.     The introduction section is inapplicable. Need to change the introduction considerably. Try to include the existing research limitations also, how the present research unravels those limits.

3.     L40 and 44: put proper references

4.     L77, 141, and so on: Sources (manufacturer name, city, country) need to mention all the chemicals, reagents, and equipment used in this manuscript.

5.     L141-144: What were the criteria for the selection of antibiotics? Verify your statement with proper references.

6.     L161: “in vivo” should be italic

7.     L232: P should be capitalized and italic “P < 0.05”. Make corrections throughout the manuscript.

8.     L247: “L. rhamnosus” bacteria name should be italic

9.     Tables 2 and 3: Many spacing and punctuation marks problems are found. Check

10.  Table 4: Were the analysis performed in triplicate? Authors should add the statistical analysis result.

11.  The conclusion needs to address future perspectives.

  1. Many spacing and punctuation marks problems are found throughout the manuscript. Revision required.

Author Response

Reviewer 3

Comments and Suggestions for Authors

The manuscript "Probiotic bacteria from human milk can alleviate oral’s bovine casein sensitization in juvenile Wistar rats" has an interesting result. However, for publication in Microorganisms, the manuscript needs to be improved.

The discussion needs to review and compare the data with the literature.

This has been modified accordingly in the revised version

Many grammatically problematic sentences were found throughout the manuscript, which must be checked and corrected precisely.

This has been corrected in the revised version

  1. L15: Use the full form of all abbreviated words for the first time throughout the manuscript

This has been corrected in the revised version

  1. The introduction section is inapplicable. Need to change the introduction considerably. Try to include the existing research limitations also, how the present research unravels those limits.

We modified the introduction to make it more appropriate. We hope the new version of the introduction will satisfy Reviewer 3

  1. L40 and 44: put proper references

This has been corrected in the revised version

  1. L77, 141, and so on: Sources (manufacturer name, city, country) need to mention all the chemicals, reagents, and equipment used in this manuscript.

This has been corrected in the revised version

  1. L141-144: What were the criteria for the selection of antibiotics? Verify your statement with proper references:

Literature reported that probiotics showed resistance toward different classes of antibiotics including glycopeptides (i.e, vancomycin), aminoglycosides (i.e, streptomycin and gentamicin), mono-bactams (i.e, aztreonam) and fluoroquinolones (i.e, ciprofloxacin) all of which are broad spectrum antibiotics known to be effective against Gram positive and Gram negative bacteria.

In fact, we tested 15 antibiotics (only 11/15 are presented in the manuscript). The antibiotic susceptibility test was performed with different antibiotics which are the most commonly used in clinics.

We modified the manuscript accordingly and used proper references in the revised version.

  1. L161: “in vivo” should be italic

This has been corrected in the revised version

  1. L232: P should be capitalized and italic “P< 0.05”. Make corrections throughout the manuscript.

This has been corrected in the revised version

  1. L247: “L. rhamnosus” bacteria name should be italic

This has been corrected in the revised version

  1. Tables 2 and 3: Many spacing and punctuation marks problems are found. Check

This has been corrected in the revised version

  1. Table 4: Were the analysis performed in triplicate? Authors should add the statistical analysis result.

This has been corrected in the revised version

  1. The conclusion needs to address future perspectives.

Conclusion has been modified accordingly in the revised version

  1. Many spacing and punctuation marks problems are found throughout the manuscript. Revision required.

This has been corrected in the revised version

Round 2

Reviewer 1 Report

The authors answer major issues but still, some clarification should be addressed before being accepted.

1- the hypothesis of the current study should be revised to be clear.

2- the scientific names must be in italics, for instance, see lines 274, 275, 391, 527, 543, etc...

3- The figures or tables should be placed after they are first mentioned.

 4- In figure 2, the phylogenetic tree should be reconstructed to be clear.

Author Response

1- the hypothesis of the current study should be revised to be clear.

We add the following sentence to make our hypothesis more clear: "

We hypothesized that prevention of allergy using probiotics may be more effective at juvenile age and that bacterial strains from human milk origin should be the most appropriate probiotics for that purpose."

2- the scientific names must be in italics, for instance, see lines 274, 275, 391, 527, 543, etc...

This has been corrected in the revised version

3- The figures or tables should be placed after they are first mentioned.

This has been corrected in the revised version

 4- In figure 2, the phylogenetic tree should be reconstructed to be clear.

This has been corrected in the revised version